# Exploring Food Access and Sociodemographic Correlates of Food Consumption and Food Insecurity in Zanzibari Households

**DOI:** 10.3390/ijerph16091557

**Published:** 2019-05-04

**Authors:** Maria Adam Nyangasa, Christoph Buck, Soerge Kelm, Mohammed Sheikh, Antje Hebestreit

**Affiliations:** 1Leibniz Institute for Prevention Research and Epidemiology—BIPS, 28359 Bremen, Germany; nyangasa@leibniz-bips.de (M.A.N.); buck@leibniz-bips.de (C.B.); 2Center for Biomolecular Interactions Bremen, University Bremen, 28334 Bremen, Germany; kelm@uni-bremen.de; 3Environmental Analytical Chemistry and Eco-toxicology, Zanzibar State University, P.O. Box 146 Unguja, Zanzibar; m.sheikh@suza.ac.tz

**Keywords:** demographic correlates, food access, household, food insecurity experience scale, Zanzibar, sub-Saharan Africa

## Abstract

Rapid growth of the Zanzibari population and urbanization are expected to impact food insecurity and malnutrition in Zanzibar. This study explored the relationship between food access (FA) and sociodemographic correlates with food consumption score and food insecurity experience scale. Based on cross-sectional data of 196 randomly selected households, we first investigated the association between sociodemographic correlates and Food Consumption Score (FCS) and Food Insecurity Experience Scale using multilevel Poisson regression. Secondly, the role of FA in these associations was investigated by interaction with the respective correlates. About 65% of households had poor food consumption, and 32% were severely food-insecure. Poor FA was more prevalent in households with poor food consumption (71%). Polygamous households and larger households had a higher chance for severe food insecurity. In the interaction with FA, only larger households with poor FA showed a higher chance for severe food insecurity. In households having no vehicle, good FA increased the chance of having acceptable FCS compared to poor FA. By contrast, urban households with good FA had a twofold chance of acceptable FCS compared to rural household with poor FA. Poor FA, poor food consumption and food insecurity are challenging; hence, facilitating households’ FA may improve the population’s nutrition situation.

## 1. Introduction

The world population is expected to increase by 2.5 billion between 2007 and 2050, with most of the growth foreseen to occur in urban areas of developing countries [1]. This rapid growth and urbanization are expected to increase poverty and negatively impact the food security environment of urban dwellers, leading to food insecurity and malnutrition [2]. According to the World Bank report from 2015 [3], 29% of Tanzanians could not meet their basic consumption needs, and about 10% of the population could not afford to buy basic food stuff. In Tanzania and other developing countries, the leading factor in household food insecurity in urban areas is the dependency on food purchase [2,4,5]. Therefore, a slight increase in food prices has a major impact on vulnerable households, pushing them into hunger and poverty [6]. About 80% of the household food requirement in peri-urban areas of Unguja Island, Zanzibar is purchased [7,8], while about 60% of food consumed in rural areas is obtained through home gardening and farming [9]. Home gardening and farming contribute substantially to households’ own food production [10] and may thus enhance household food availability and food security in rural areas. Own food production increases purchasing power, due to savings on food bills [4,11] and income from selling of the produce. It also provides a diversity of nutritious food that helps to improve the health status of the household [11] as well as serves as a means of food provision during food shortage.

Studies conducted in Nigeria [12] and Ethiopia [13] have shown that male heads of household (HH) play a substantial role in determining household food security, while others reported that female HH are more likely to spend most of their income on food, thus guaranteeing food security for their households [14,15]. Household size has also been shown to influence food security and acceptable food consumption, with smaller household size being associated with household food security [16] and acceptable food consumption, and large household size with poor food consumption and perceived food insecurity [17]. Food insecurity has also been found to be associated with low monthly income [18]. Income earned from any source improves the food situation of a household [19]; thus, households with more employed adult members are likely to have a better food situation compared to households with more unemployed adult members [20].

Several studies conducted in low-income countries, especially in Africa [4,5,16,18], have investigated the determinants of either household food security or food consumption behavior in households. However, this is the first study in Zanzibar that recruited randomly selected households and enrolled all members of a household for further insights on correlates of food insecurity and food consumption. The present study aimed to explore the relationship between food access (FA) and sociodemographic household factors with the Food Consumption Score (FCS) and Food Insecurity Experience Scale (FIES). Findings from this study can provide baseline information on the interaction between food access and household factors, and the collected data can be used for further research on health interventions to improve food consumption and food security in Zanzibar.

## 2. Materials and Methods

### 2.1. Study Area, Population, and Sampling

A population-based cross-sectional study was conducted in 2013 in Unguja Island, Zanzibar, whereby entire households were enrolled as sampling units. For the purposes of this study, a household is defined according to Beaman and Dillon [21], with emphasis on eating from the same pot, i.e., having the same food provider. This is particularly important as our study population consisted not only of monogamous but also of polygamous families, who do not necessarily live together in the same house or compound.

Household aspects were reported by the head of household in a questionnaire-administered personal interview. In total, 244 randomly selected households were contacted from 80 Shehias (wards), and 239 (97.9%) participated in the survey. Due to missing information on socioeconomic status, demographic correlates, and responses from FCS and FIES instruments, 43 households were excluded from the analysis, resulting in a final sample of 196 (82%) households. Further details on sampling procedures, data collection, and quality management are provided elsewhere [22].

Prior to the data collection process, all participants gave written, informed consent. All procedures applied in this study were approved by the Ethics Committees of the University of Bremen (in September 26, 2013) as well as the Zanzibar Ministry of Health and the Zanzibar Medical Research and Ethics Committee (ZAMREC/0001/AUGUST/013) in accordance with the ethical standards according to the 1964 Declaration of Helsinki and its later amendments.

### 2.2. Questionnaires

A structured household questionnaire was used to collect general household information. The information included data on socioeconomic and demographic indicators of the household and of the head of household, such as area of residence, number of animals owned, number of vehicles belonging to the household, household size, marital status, education level of the HH, occupation of the HH, etc. In cases of polygamy, household information was also collected from the households of the other wife or wives. Information on indicators of household food consumption was collected using a standardized questionnaire (Food Consumption Score, FCS), adapted from the United Nations World Food Programme (UNWFP) [23]. Household food insecurity was measured at the household level using a standardized questionnaire for the Food Insecurity Experience Scale (FIES), which was adapted from the Food and Agriculture Organization (FAO) [24]. All questionnaires were developed in English, translated into Swahili and back-translated to check for translation errors.

#### 2.2.1. Food Consumption Score (FCS)

FCS is a composite score constructed from (1) household dietary diversity based on nine food groups (staples, pulses, fruits, vegetables, meat and fish, dairies, sugar, oil and fat, condiments) consumed during the 7 days preceding the survey, (2) food frequency, counted as the number of days each food group was consumed during the 7 days preceding the survey, and (3) relative nutritional importance of different food groups, applying a weighting system [23], thus reflecting the quality and quantity of food consumed in the household.

Higher weights were given to energy-dense foods with proteins of high quality and a range of bioavailable micronutrients, while lower weights were given to oil and sugar, which are energy-dense but contain—if any—proteins of low quality and low levels of micronutrients [25]. Cut-off points established by the UNWFP [23] were used to classify FCS. They were computed by summing up the weighted frequencies of the different food groups consumed in the household. FCS ≤ 28 was categorized as “poor”, FCS > 28 and < 42 as “borderline”, and FCS ≥ 42 as “acceptable” [23]. For the regression analysis in this study, poor consumption and borderline were merged to a new category, “poor”, resulting in two categories of food consumption score, i.e., “poor” and “acceptable”.

#### 2.2.2. Food Insecurity Experience Scale (FIES)

Food insecurity was assessed using the Food Insecurity Experience Scale (FIES), a standardized set of questions developed by the FAO [24] that has been applied in a large number of countries following a standardized procedure. The scale, which is an experience-based metric of severity of food insecurity that relies on people’s direct/actual responses, includes components of uncertainty and worry about food, inadequate food quality, and insufficient food quantity. It consists of a set of eight items that assess food-related behaviors associated with difficulties in accessing food due to limited resources. The instrument measures the degree of food insecurity/hunger experienced by individuals during the 12 months preceding the survey.

Household scores of food insecurity on the eight items were scaled based on a Rasch model as an application of the item–response theory (IRT) [26]. In IRT, the response to each item is modelled as a function of item and household parameters to measure the position of households on a latent trait, independently of the item difficulty. We conducted the Rasch model using the eRm package 0.16–2 [27] in R 3.5.1 to derive household scores for the following regression analyses. Households who responded ‘no’ to any of the eight items received the lowest value. Item characteristic curves indicated item reliability, and the person separation index was found to be high (0.84). Household scores ranged from −3.57 to 3.76 and clustered either to the lowest negative scores (<−2), around 0 (−2 to 2) or to the highest scores. Based on these cut-off points, we categorized household scores into mild, moderate, and severe (hunger), which are the three categories used by the FAO [24] to define levels of food insecurity/hunger. For the Poisson regression analysis, food security, mild and moderate food insecurity were dichotomized into “mild to moderate” (0) and “severe food insecurity” (1). A household is considered as food-secure if members have always had enough food and no hunger worries [24]. 

### 2.3. Correlates

Correlates of FCS and FIES were assessed for the HH (gender, education level, number of jobs, marital status) and at household level (household size, number of types of animals kept in the household, vehicles owned by any of the household members, area of residence, and food access). The highest education level of the HH was assessed using the International Standard Classification of Education (ISCED) [28] and was categorized for the analysis as low education level (primary school and below) and high education level (secondary school and above). Number of jobs of HH for assessing main household income was defined as “no job” and “≥1 job”. Marital status of HH was calculated in three categories: Married monogamous, married polygamous, and other (single, widowed, cohabitating or divorced). To facilitate interpretation, two categories for marital status were derived: Married (monogamous or polygamous) and not married (single, widow, cohabitation, divorced). Cohabitation was categorized as “not married” since it is characterized by a different socioeconomic status compared to those households with married HH. Using the mean household size of 6 members in this study population as a cut-off, household size was classified as large (six or more members) and small (less than six members). To assess household wealth, the number of types of animals kept in the household from a list of six items including ducks, goats, sheep, cows, fish, and chicken, and the number of assets from a list of eight items including electricity, radio, mobile phone, iron, kerosene lamps, television, refrigerator, and non-mobile phone was summed up. The median number of animals and assets was five, and this figure was used to categorize wealth into wealthy (≥5) and poor (<5). The number of vehicles per household was assessed as type of transportation owned by any member of the household from a list of 5 categories (bicycle, car/truck, boat, motorcycle, none). Two categories were built: “none” and “at least one type of vehicle”. Area of residence was assessed as rural or urban. 

An important component of food consumption and food insecurity included in the analysis was FA. In this study, FA was defined as the ability of a household to acquire adequate amount of food through mixed strategies. Indicators for FA were assessed, and derived variables were combined to a composite score (see Table 1 below). The derived variables were: (a) Food source; main source of food consumed during the last seven days (purchased, borrowed, own production, traded food/barter, received as gift, food aid, other), (b) food purchased; types of food (cereals, starchy vegetables/ tubers, vegetables, fruits, legumes, meat, egg, milk, fish, oils and fats, any kind of beverage, other), frequently bought from shop/market during the last seven days, (c) own food; types of foods (from those listed above) of own household production, and (d) market distance; distance to the nearest market or shop (<30 min, 30 to 60 min, 1 h to 2 h, >2 h).

To classify households as having poor or good FA, a composite score for FA was computed as the mean of the values of the derived variables multiplied by 4 (number of all derived variables). The FA-score ranged from 4–8, and FA was then categorized as “poor food access” (≤6) and “good food access” (>6). The cut-off was set according to the distribution of FA-status in the study population, as half of the population included in the study had FA score of 6 and below. 

### 2.4. Statistical Analysis

Study characteristics such as socioeconomic and demographic variables were calculated for categories of food consumption and food insecurity. First, the associations between the exposure variables (food access, socioeconomic and demographic correlates) with either food consumption (FCS; Model 1) or food insecurity (FIES; Model 2) as outcome variables were explored. To explore these associations, linear, Poisson, and logistic regression models were considered and evaluated with regard to model fit and Pearson residuals as well as quantile-quantile (Q–Q) plots. Eventually, we conducted multilevel logistic regression models to calculate odds ratios (OR) and 95% confidence limits (CI) and to account for clustering within Shehia level using a random intercept. Secondly, statistical interactions between each correlate and food access (correlate*FA) were investigated. Hence, in Models 3a–h and 4a–h, the predictive power of each socioeconomic and demographic factor with FA on FCS and FIES was tested separately. Each model was again adjusted for the remaining correlates, including a random intercept for the Shehia level. All statistical analyses were performed using SAS 9.3 (SAS Institute, Cary, NC, USA). Due to the exploratory design of our study, we only considered confidence limits as a precision measure of the point estimates but did not apply a level of significance. Moreover, we did not adjust for multiple testing. Noteworthy associations are presented considering higher (OR > 1.5) or lower (OR < 0.66) chances for the modelled response category.

## 3. Results

### 3.1. Household Characteristics 

The sample data were based on the responses of the HH. The majority of the households were headed by men (63%, 123/196), and about 55% of the HH (107/196) were in a monogamous marriage. More than half of the HH had one or more sources of income that he/she contributed to the household. Most of the households were in rural areas, and the overall mean household size of the participating households was 6 persons. More than 60% of the households had a good socioeconomic status, with one or more than one animal kept and at least one vehicle owned by a member belonging to the household. Overall, about 65% of the households had poor food consumption, and about 32% were severely food-insecure (Table 2). Acceptable food consumption was more prevalent in households with higher-educated HH (40%), in monogamous households (38%), and in larger households (six or more members) (38%). Severe food insecurity was more prevalent in polygamous households (40%), in households with low-educated HH (40%) and in larger households (six or more members) (40%). Looking at each question of the FIES, the majority of the households (73.5%) indicated having eaten few kinds of food in the last 12 months due to lack of money, and 26% went without eating for a whole day due to lack of money (Table 3). About 54% (106/196) of the study population had poor FA, of which 71% had poor food consumption and 35% experienced severe food insecurity. 

### 3.2. Correlates of Food Consumption and Food Insecurity

Households with a higher-educated HH had a lower chance of reporting severe food insecurity (OR 0.53; 95% CI 0.28–1.08) compared to those with lower-educated HH (Table 4). Those HH married in monogamy had a higher chance of reporting acceptable food consumption (OR 1.71; 95% CI 0.55–5.38) but at the same time a higher chance of severe food insecurity (OR 1.83; 95%CI 0.55; 6.08) compared to those HH not married (single, widowed, cohabitating or divorced) (Table 4). Polygamous households had a higher chance of severe food insecurity (OR 3.95; 95% CI 1.17–13.4) and also reported higher chance for acceptable food consumption (OR 1.78; 95% CI 0.54–5.83) compared to those not married. Larger households had a higher chance of severe food insecurity (OR 2.44; 95% CI 1.16–5.13) than smaller households, while wealthy households had a lower chance of severe food insecurity (OR 0.52; 95% CI 0.25–1.11) than poor households. 

### 3.3. Role of Food Access on the Correlates, Food Consumption and Food Insecurity

There were few relevant changes in the chances observed in the interaction of food access with gender, marital status of HH, number of jobs, and wealth on both food consumption and food insecurity. However, urban households with good FA showed a higher chance of acceptable food consumption compared to rural households with poor FA (Table 5, Model 3e). Households with no vehicle had a higher chance of acceptable food consumption if they had good FA compared to those with poor FA (OR 6.21; 95% CI 1.20–32.3). However, having at least one vehicle tentatively increased the chance of having acceptable food consumption for both good and poor FA (OR 2.17; 95% CI 0.68–6.87; OR 1.83; 95% CI 0.59–5.71, respectively) (Model 3h).

In comparison to households with low-educated HH and poor FA, we observed a lower chance of severe food insecurity in households with higher-educated HH either with good FA (OR 0.42; 95% CI 0.15–1.17) or poor FA (OR 0.40; 95% CI 0.16; 1.02) and in households with lower-educated HH and good FA (OR 0.49; 95% CI 0.18–1.30) (Model 4c). Considering poor FA, larger households (six or more members) had a higher chance of severe food insecurity (OR 3.42; 95% CI 1.29–9.10) compared to smaller households (Model 4f). 

## 4. Discussion

This exploratory study aimed at adding to the on-going debate on how good FA may help to improve the nutrition situation of the Zanzibar population. We observed that poor FA, poor food consumption, and food insecurity are a problem in many Zanzibari households. Poor FA was more prevalent in households with poor food consumption and severe food insecurity. In particular, polygamous households and larger households had a higher chance of severe food insecurity. Good FA increased the chance of acceptable food consumption for urban households and households with no vehicle, whereas poor FA increased the chance for severe food insecurity for larger households. 

### 4.1. Proportions of FCS and FIES in the Study Population

The proportion of households with acceptable food consumption was about 35% in the present study. This is lower than reported in Ethiopia (73%) [29] and in the Nyarugusa refugee camp in Tanzania (86%) with a sample of 343 households in a WFP/United Nations High Commissioner for Refugees (UNHCR) study [30]. Food consumption was assessed using comparable instruments in the present study and in the UN World Food Programme Study [23]; thus, the higher proportions in the Nyarugusa camp could have been due to the fact that 83% of the households’ main source of food was from food aid, unlike in Unguja Island, where households relied mostly on purchase and on their own production. The overall proportion of mild to moderate food insecurity in our study population was about 68%, slightly higher than the 49% observed in Burkina Faso among 330 surveyed households [31], and doubles that in Ethiopia (34%) [32]. This survey was conducted during the short-rainy season (October–December 2013), which is a time for sowing and growing of food products and when most household stocks are depleted [33]. This may have contributed to the relatively low proportions of households with acceptable food consumption. Still, the proportion of households with experienced mild to moderate food insecurity was higher compared to other household surveys, which were also conducted during the lean seasons [29,31,32,34]. More than three quarters of the households in our study resided in the rural area and hence depended mostly on own food production, and food products grown in the rainy season are harvested between January and February. A survey conducted in Burkina Faso confirmed the seasonal effects on food security of 1056 households with a lower food security during the lean season compared to the post-harvest season [34]. Hence, data collection should also be conducted during the post-harvest season in order to gain insight into FCS and FIES when food availability improves on Unguja Island.

### 4.2. Role of Food Access on the Correlates, Food Consumption and Food Insecurity

In the present study, higher education level of the HH seemed to influence the level of food consumption and food insecurity of the households with poor FA, agreeing with findings from Rwanda [35] and Uganda [36], reflecting that “some level of education is important to household food security”. It is assumed that a literate HH has a greater capacity of adapting to improved technologies and coping strategies, thus increasing production/food supply in the household [16]. The lower chance for severe food insecurity in households with higher-educated HH and good FA, higher-educated HH and poor FA, and lower-educated HH and good FA indicated that education level of the HH as well as FA both played a comparable role in impacting the food insecurity status of the household. 

The fact that in this study, larger households had higher chances for severe food insecurity than smaller households may be explained by the fact that smaller household sizes are generally better manageable in terms of food demand and supply. The latter can be improved through own production, which increases food consumption and decreases food insecurity as also reported in other studies [18,37]. In contrast to our findings, studies from Niger and Nigeria observed a decreasing likelihood of a household being food-insecure with an increasing household size [38]. However, to estimate the effect of household size on food insecurity is challenging, as this would require verifying numerous other factors such as the number of all active members in the household (contributors of income or food), income sources (salary), and expenditure on food. While the number of jobs of the HH showed no effect on food insecurity, wealthier households, on the other hand, revealed a lower chance of having severe food insecurity compared to poor households. Interestingly, our results showed that larger households with poor FA had a higher chance for severe food insecurity compared to smaller households, while those with good FA only had a smaller chance for severe food insecurity compared to small households with poor FA, this further confirms the role FA played in impacting food insecurity in the study population. Studies investigating the role of FA on food consumption and food insecurity in Sub-Saharan Africa are scarce, and more research is advisable.

In our study, households with polygamous HH had a higher chance for of severe food insecurity than households with not married (e.g., widowed or divorced) HH. While this is in line with previous findings from Tanzania mainland [39], other studies reported contrasting findings [40,41]. The latter postulated that the large number of individuals in polygamous households means that a great number of individuals can provide financial and labor support among each other, thus reducing the chance of having food insecurity. However, these studies compared polygamous households against monogamous households. When we compared monogamous against unmarried households, we observed a relatively lower chance of experiencing severe food insecurity. This is supporting our findings that larger households were more likely to be food-insecure compared to smaller households. 

The finding that urban households with good FA experienced a higher chance of acceptable food consumption than rural households with poor FA may be explained by the good infrastructure in urban areas, which enables good accessibility for foods and may potentially enhance the food diversity of households. The good infrastructure includes factors such as the quality of roads or market density, which facilitate food distribution and transport into the communities. Studies in Malawi and Kenya [42,43] also reported that food insecurity for rural households was affected by long distance to the market and poor market access. Furthermore, in Malawi, the distance to the market was reported to affect food consumption for both rural and urban households [42]. The fact that in our study, even households without a vehicle had a higher chance of acceptable food consumption when they had good FA compared to those with poor FA indicates the importance of other aspects of the FA—in addition to infrastructure—such as borrowing foods, receiving foods as a gift, and own food production or animal rearing. The interplay of multiple factors facilitating food accessibility should be considered in future intervention studies. 

### 4.3. Strengths and Limitations

This exploratory study provides valuable data on the interplay of food access, socioeconomic and demographic correlates with food consumption and food insecurity of Zanzibari households. Even though our study was conducted during the short-rainy season, which made it difficult to have access to some of the villages, an important strength was the overall high proportion of households that participated in the study. Further, the highly standardized study protocol using partly validated and pretested methods and instruments is a clear strength of the study. When comparing the study characteristics such as HH demography (gender, marital status, education level, occupation), household demography (rural/urban, household size), and the socioeconomic status of the household of the full survey sample and the sample presented in this study sample, no substantial differences were observed (results not shown). Thus, a selection bias can be ruled out. 

The study, however, has limitations. Firstly, the FIES was explicitly developed for cross-cultural comparability and assesses correlates of food insecurity across different areas that have the same climatic or agricultural calendar [44]. We, however, conducted our study only in Unguja Island and hence have no basis for a comparison with other Zanzibari islands such as Pemba Islands, or with the population of Mainland Tanzania. As household information was based on self-reports, social desirability could have influenced the responses given. The survey was conducted during October–December, which is the time of the year when most household stocks are depleted (lean season) [33]; this may have affected our results on the food consumption and food insecurity situation of the households and must be acknowledged as a limitation. Further, the collection of cross-sectional data means that effects of seasonal variations could not be investigated. To overcome this limitation, it would be advisable to collect longitudinal data.

One major limitation we like to address is that the overall study was planned, powered, and conducted to estimate the prevalence of malnutrition in the Zanzibari population [22]. However, with this study, we intended to explore important household survey data on a broader level even though for this particular approach, the sample size was underpowered. Nevertheless, the data are a useful source for exploring the role of FA in the association between sociodemographic household factors with food consumption and food insecurity. Findings from this study will add knowledge and inform the development of intervention strategies and policies aiming at improving food consumption and food security in Zanzibar. Still, more research with a larger sample size is advisable.

## 5. Conclusions

Based on our findings, poor access to food may be seen as a modifiable factor for food consumption and perceived food insecurity in Zanzibari households, in particular for the association with educational level and household size. To improve food and nutrition security in Zanzibar, implementation of policies and programs that address education activities and different forms of practical coping strategies, such as efficient food storage techniques and home gardening, in their agendas are needed, particularly in rural areas. In parallel, strategies should consider improvement of infrastructure to facilitate distribution of produce within the rural–urban areas, as well as education campaigns on food quality and utilization, emphasizing on the importance of food group and balanced diets.

## Figures and Tables

**Table 1 ijerph-16-01557-t001:** Overview of measured indicators for FA and derivation of variables for the composite score.

Indicator of Food Access	Derived Variable	Categories
**Main source of food consumed** One main source to be selected from 6 categories: purchased, borrowed, traded food/barter, received as gift, food aid, own production	Food source = one main source per household	1: borrowed, received as gift, food aid, other 2: own production, traded food/barter, purchased
**Types of food groups frequently bought from shop/market** Food group (e.g., cereals) out of 11 food groups was 0: not bought, 1: bought	Food purchased = sum of all food groups purchased	1: ≤4 food groups, 2: >4 food groups
**Types of food groups of own household production** Food groups out of 11 food groups 0: not produced (purchased, borrowed, traded food/barter, received as gift, food aid), 1: own production	Own food = sum of all food groups with own production	1: ≤2 food groups, 2: >2 food groups
**Distance to the nearest market/shop** far (>30 min walking distance); near (<30 min walking distance)	Market distance	1: far, 2: near

**Table 2 ijerph-16-01557-t002:** Proportion of food consumption and food insecurity experience scale according to demographic and socioeconomic factors.

	Food Consumption Score				Food Insecurity Experience Scale				Total
	Poor		Acceptable		Mild to Moderate		Severe		
	*N*	%	*N*	%	*N*	%	*N*	%	*N*
All	128	65.3	68	34.7	134	68.4	62	31.6	196
**Household Demographics**									
**Gender**									
Male	72	58.5	51	41.5	80	65.0	43	35.0	123
Female	56	76.7	17	23.3	54	74.0	19	26.0	73
**Marital status of HH**									
Not married ^a^	27	79.4	7	20.6	28	82.4	6	17.6	34
Married monogamous	66	61.7	41	38.3	73	68.2	34	31.8	107
Married polygamous	35	63.6	20	36.4	33	60.0	22	40.0	55
**Education level**									
Low	69	71.1	28	28.9	59	60.8	38	39.2	97
High	59	59.6	40	40.4	75	75.8	24	24.2	99
**Number of jobs**									
No job	61	70.1	26	29.9	57	65.5	30	34.5	87
One or more jobs	67	61.5	42	38.5	75	68.8	24	22.2	109
**Area**									
Rural	105	68.6	48	31.4	102	66.7	51	33.3	153
Urban	23	53.5	20	46.5	32	74.4	11	25.6	43
**Household size** ^b^									
Small	68	68.7	31	31.3	75	75.8	24	24.2	99
Large	60	61.9	37	38.1	59	60.8	38	39.2	97
**Socioeconomic Factors**									
**Wealth (household assets and animals)** ^c^									
Poor	61	73.5	22	26.5	51	61.4	32	38.6	83
Wealthy	67	59.3	46	40.7	83	73.5	30	26.5	113
**Number of vehicles**									
None	31	72.1	12	27.9	28	65.1	15	34.9	43
At least one vehicle	97	63.4	56	36.6	106	69.3	47	30.7	153
**Food access** ^d^									
Poor	75	70.8	31	29.2	69	65.1	37	34.9	106
Good	53	58.9	37	41.1	65	72.2	25	27.8	90

^a^ Not married includes single, divorced, widow, and cohabitation; ^b^ cut-off was derived from the mean number of household members in this study, small (≤6) and large (>6); ^c^ wealth (poor <5 and wealthy ≥5, calculated as the median number of animals and assets in the household); ^d^ cut-off (poor ≤6, good >6).

**Table 3 ijerph-16-01557-t003:** Questions of the Food Insecurity Experience Scale and affirmatively answered questions by the study population in Zanzibar (*N* = 196). During the last 12 months, was there a time when…

No	Food Insecurity Experience Scale Questions	*N*	%
1	You were worried you would run out of food because of a lack of money?	112	57.1
2	You were unable to eat healthy and nutritious food because of a lack of money?	134	68.4
3	You ate only a few kinds of foods because of a lack of money?	144	73.5
4	You had to skip a meal because there was not enough money to get food?	100	51.0
5	You ate less than you thought you should because of a lack of money?	117	59.7
6	Your household ran out of food because of a lack of money?	103	52.6
7	You were hungry but did not eat because there was not enough money for food?	76	38.8
8	You went without eating for a whole day because of a lack of money?	51	26.0

**Table 4 ijerph-16-01557-t004:** Associations of socioeconomic and demographic correlates of 196 households with food consumption (Model 1) and food insecurity (Model 2) in terms of odds ratios (OR) and 95% confidence intervals (CI) as well as model fit (generalized chi-square/degrees of freedom), respectively.

	Model 1: Food Consumption (Ref: Poor)		Model 2: Food Insecurity (Ref: Mild to Moderate)	
**Model fit**	$\chi^{2}/DF$	0.87	$\chi^{2}/DF$	0.86
Between Shehia variance (SE)		0.48 (0.38)		0.37 (0.38)
	**OR**	**(95% CI)**	**RR**	**(95% CI)**
Gender (ref: female)	1.76	(0.75–4.11)	1.65	(0.68–4.01)
**Marital status of HH (ref: not married)**				
monogamous	1.71	(0.55–5.38)	1.83	(0.055–6.08)
polygamous	1.78	(0.54–5.83)	3.95	(1.17–13.4)
Education (ref: low)	1.36	(0.69–2.70)	0.53	(0.26–1.08)
Number of jobs (ref: no job)	1.22	(0.59–2.50)	0.58	(0.28–1.22)
Area of residence (ref: rural)	2.08	(0.85–5.10)	0.64	(0.24–1.70)
Household size (ref: small)	1.02	(0.51–2.05)	2.44	(1.16–5.13)
Wealth (ref: poor ≥ 5)	1.35	(0.64–2.83)	0.52	(0.25–1.11)
Number of vehicles (ref: none)	1.04	(0.42–2.57)	0.78	(0.32–1.89)
Food access (ref: poor ≤ 6)	1.56	(0.79–3.10)	0.69	(0.33–1.43)

**Table 5 ijerph-16-01557-t005:** Results of the multilevel logistic regressions in terms of odds ratios (OR) and 95% confidence limits as well as model fit (generalized chi-square/degrees of freedom) to investigate the interaction of food access with socioeconomic and demographic correlates on food consumption (Model 3a–h) and food insecurity (Model 4a–h), each adjusted for the remaining correlates.

Model	Covariate	Food Access	Model 3: Food Consumption (Ref: Poor)					Model 4: Food Insecurity (Ref: Mild to Moderate)				
			Ref: *N* (%)	OR	(95% CI)	Between Shehia Variance (SE)	$\chi^{2}/DF$	Ref: *N* (%)	OR	(95%CI)	Between Shehia Variance (SE)	$\chi^{2}/DF$
a	**Gender**											
	Male	Good access	34 (55.7)	3.03	(0.96–9.62)	0.48 (0.38)	0.88	41 (67.2)	0.85	(0.29–2.53)	0.41 (0.39)	0.87
	Male	Poor access	38 (61.3)	2.26	(0.73–7.00)			39 (62.9)	0.99	(0.36–2.78)		
	Female	Good access	19 (65.5)	2.24	(0.66–7.66)			24 (82.8)	0.44	(0.12–1.61)		
	Female	Poor access	37 (84.1)	1.00				30 (68.2)	1.00			
b	**Marital status** ^a^											
	Married	Good access	43 (55.8)	1.96	(0.49–7.88)	0.50 (0.39)	0.85	53 (68.8)	1.67	(0.43–6.39)	0.39 (0.39)	0.87
	Married	Poor access	58 (68.2)	1.09	(0.28–4.25)			53 (62.4)	2.28	(0.63–8.20)		
	Not married	Good access	10 (76.9)	0.62	(0.09–4.20)			12 (92.3)	0.44	(0.04–5.01)		
	Not married	Poor access	17 (81.0)	1.00				16 (76.2)	1.00			
c	**Education**											
	High	Good access	28 (58.3)	2.21	(0.82–5.98)	0.49 (0.39)	0.87	37 (77.1)	0.42	(0.15–1.17)	0.38 (0.38)	0.88
	High	Poor access	31 (60.8)	2.09	(0.80–5.48)			38 (74.5)	0.40	(0.16–1.02)		
	Low	Good access	25 (59.5)	2.53	(0.91–7.06)			28 (66.7)	0.49	(0.18–1.30)		
	Low	Poor access	44 (80.0)	1.00				31 (56.4)	1.00			
d	**Number of jobs**											
	One or more	Good access	36 (60.0)	2.05	(0.80–5.24)	0.55 (0.40)	0.86	42 (70.0)	0.44	(0.17–1.13)	0.50 (0.42)	0.83
	One or more	Poor access	31 (63.3)	1.95	(0.73–5.17)			35 (71.4)	0.36	(0.14–0.95)		
	No Job	Good access	17 (56.7)	2.81	(0.96–8.25)			23 (76.7)	0.32	(0.10–1.01)		
	No Job	Poor access	44 (77.2)	1.00				34 (59.6)	1.00			
e	**Area of residence**											
	Urban	Good access	6 (35.3)	5.48	(1.42–21.2)	0.46 (0.38)	0.88	14 (82.4)	0.44	(0.09–2.08)	0.39 (0.38)	0.87
	Urban	Poor access	17 (65.4)	1.21	(0.39–3.71)			18 (69.2)	0.62	(0.19–1.96)		
	Rural	Good access	47 (64.4)	1.16	(0.53–2.51)			51 (69.9)	0.70	(0.32–1.53)		
	Rural	Poor access	58 (72.5)	1.00				51 (63.8)	1.00			
f	**Household size**											
	Large	Good access	23 (52.3)	1.56	(0.59–4.12)	0.46 (0.38)	0.88	30 (68.2)	1.56	(0.54–4.48)	0.38 (0.38)	0.87
	Large	Poor access	37 (69.8)	0.70	(0.26–1.86)			29 (54.7)	3.42	(1.29–9.10)		
	Small	Good access	30 (65.2)	1.08	(0.41–2.84)			35 (76.1)	1.15	(0.41–3.22)		
	Small	Poor access	38 (71.1)	1.00				40 (75.5)	1.00			
g	**Wealth**											
	Wealthy	Good access	28 (53.8)	2.21	(0.78–6.28)	0.48 (0.38)	0.87	40 (76.9)	0.38	(0.13–1.09)	0.43 (0.40)	0.86
	Wealthy	Poor access	39 (63.9)	1.59	(0.57–4.48)			43 (70.5)	0.42	(0.16–1.14)		
	Poor	Good access	25 (65.8)	1.90	(0.63–5.71)			25 (65.8)	0.51	(0.18–1.47)		
	Poor	Poor access	36 (80.0)	1.00				26 (57.8)	1.00			
h	**Number of Vehicles**											
	At least one	Good access	48 (60.8)	2.17	(0.68–6.87)	0.44 (0.38)	0.89	56 (70.9)	0.51	(0.18–1.43)	0.38 (0.38)	0.86
	At least one	Poor access	49 (66.2)	1.83	(0.59–5.71)			50 (67.6	0.63	(0.24–1.69)		
	None	Good access	5 (45.5)	6.21	(1.20–32.3)			9 (81.8)	0.31	(0.05–2.06)		
	None	Poor access	26 (81.3)	1.00				19 (59.4)	1.00			

^a^ Two categories for marital status of HH were used (1 = not married (single, widow, divorce, cohabitation) 2 = married (monogamous or polygamous)).

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
