# Peer review of "Exploring Food Access and Sociodemographic Correlates of Food Consumption and Food Insecurity in Zanzibari Households"

_ijerph, 2019, doi:10.3390/ijerph16091557_

Round 1

Reviewer 1 Report

A key problem with the manuscript remains the over-interpretation of non-statistically significant finding. A change in the text where the level of significance from the original manuscript “statistical significance was set at α = 0.05.” has become “Due to the exploratory design of our study, significance threshold was set to α = 0.1.” simply cannot be justified. The level of significance used cannot be changed after the statistical results have been seen.

First, presenting 90% confidence intervals and using a two-sided alpha of 0.10 does not, in my opinion as a biostatistician, reflect an exploratory study; it simply lowers the bar for apparent statistical significance to an unconventionally low level. If it is ever acceptable, and, again as a biostatistician, I would argue that it almost absolutely never is, it would need to have been part of the a priori protocol or data analysis plan before any statistical modelling was performed, at the very least in order to avoid a reader wondering if this value was not chosen to make almost statistically significant findings appear to be significant. Much more importantly, in the original manuscript, you stated that “All statistical analyses were performed using SAS 9.3 (SAS Institute, Cary, NC, U SA) and statistical significance was set at α = 0.05.” and so you absolutely cannot change the level of significance now.

One of my main comments about the original version of the manuscript was that non-statistically significant findings were being misleadingly described as if they were statistically significant. The lack of adjustment for multiplicity (also mentioned in my original review) means that the study-wise type 1 error rate will already be well above its nominal level, and the only concessions I would make for an exploratory study would be not adjusting for multiplicity (with this acknowledged in the limitation) and noting non-statistically significant tendencies (p<0.10) where these provide additional support to statistically significant results, form part of a persistent pattern of results, or indicate areas worth further study (i.e., not as indicating evidence in themselves). When this is done, it is important to operationally define “tendency” in the statistical methods, and explain why these have been noted. It cannot be used for clearly non-statistically significant findings. If your study was not adequately powered to detect important differences in the first place, that raises ethical concerns. On this note, the justification for the sample size appears to be missing, and please remember that this cannot be done retrospectively (it is part of the design of the study). If the study is underpowered, this will be indicated in wider than desired 95% CIs and your interpretation of the results will include discussion of whether practically important effects are contained in those intervals, suggesting that further research is needed, or not.

As mentioned last time, model diagnostics should be discussed in the statistical analysis section (from my previous review: “Model diagnostics are necessary and these should be explained in the methods.”) so that the reader is reassured that basic model checks have been performed. I would anticipate here at least a discussion of the distribution of random effects and inspection of level-1 residuals (remember to be clear which type(s) you used, e.g. Anscombe, deviance, or Pearson), but you could also look at influence diagnostics and specification tests (beyond the goodness of fit tests presented). While these diagnostics are more complex than those from a linear mixed model, they are essential for the reader’s confidence in the results.

I’m willing to accept your comments about categorising numbers of animals and vehicles, but I think you need to more carefully consider your argument about categorising food consumption, food access, and food (in)security. While presenting results to non-researchers is often easier when using categories, these results depend on the cut-points used (and medians are seldom replicable across studies, although the number of vehicles might be a rare exception there) and are, unavoidably, statistically inefficient. Note that it is not required for values to be spread out for continuous variables, although careful consideration of leverage points is always indicated. Irrespective of the distribution, using these variables as continuous measures (as long as potential non-linearities are investigated and modelled as appropriate, and care is taken with leverage points) cannot possibly introduce problems that did not also exist when looking at the same data using categories. If the model fit was poorer with a continuous versions, the only explanation would be unmodelled non-linearities or a failure to account for unusual data points that exert appreciable leverage on the model’s estimates. While categorical versions of the variables might appear to make things simpler, this approach introduces additional problems that are often not recognised (including the importance of the distribution of values within categories, which is difficult to make entirely clear in the results, when considering the interpretation of effects). Given the work you have done with the Rasch modelling of the food (in)security variable, I am happy to leave the above as suggestions for you to consider now and in your future research, but dichotomising variables will need to be discussed as a limitation.

As also mentioned in my previous review, “Note that the results in Lines 189–194 [now Lines 216–219, but also applies to Lines 214–216] all require statistical testing also.

I will also make some specific comments below. I have made relatively few comments on the results or discussion sections as these would need rewriting once the issues noted above are corrected.

Line 38: There is a spurious space between the number and percent sign.

Lines 50: I think you mean “headS of household” here.

Line 51: “great” is also a value judgment, perhaps consider “significant” or “substantial” instead?

Line 65: “…with THE food…”

Line 76: “…who DO not…”

Lines 97–98: “…at THE household level…”

Line 115: Note that these rules mean that scores of 28 are in both the “poor” and “borderline” categories and scores of 42 are both “borderline” and “acceptable”.

Line 130: Just “…application of item-response theory…” (no need for “the”)

Line 130: “IRT” not “ITR” (two instances).

Line 135: “…and THE person separation index…”

Line 138: “…to define levelS of…”

Table 1 belongs in the results section not the methods given the reporting of responses from the study. Note that the comma used as the decimal place sign in percentages should be a period.

Line 152: “…at THE household level…”

Line 162: “…living in MARRIED status…” (“marital status” is the description of the variable rather than a level of the variable)

Lines 165–166 and 167–168: If you use “including”, you don’t need parentheses around the list of items. Note that there are only six types of animals if you are counting them (“none” would be a count of 0). A similar point applies to Lines 170–171.

Lines 176–181: You don’t need the underscores in the names here and you can be more descriptive with these names. The reader doesn’t need to be concerned with the way the name is represented within the statistical software. The same point applies to Table 2.

Lines 196 and 202–203: Note the problem with 90% CIs and testing with an alpha of 0.10.

Line 210: I suggest the less ambiguous “mean” rather than “average” here (assuming that this is in fact a mean, and “median” or whatever is appropriate otherwise).

Table 3: The percentages for FIES by no job are not correct (65.5% and 39.2%).

Line 229: If you’re going to use the term “tendency”, it would need to be defined in the methods. Note that standard usage for this term is to indicate a p-value between 0.05 and 0.10 where the p-value here is clearly greater than 0.10. Results that would arise more than 10% of the time under the global null hypothesis are unlikely to be worth commenting on, particularly given the large number of hypothesis tests performed here.

Table 4: The source(s) of the effects, CIs, and p-values (which should also be shown) need to be clear in the table.

Lines 246–261: This text should indicate interaction p-values in order to justify any stratified presentations.

Author Response

Please find the responses in the attached document.

Reviewer 2 Report

Hello Authors!

Thanks for this revised version of your paper. There are still some minor issues, including some language editing... as well as one more substantial point that escaped me when I first read your paper.

The substantial point has to do with the timing of your sampling, which coincides with the vuli season, which is also the lean period. The point should be mentioned clearly. Lines 290-294 may allude to this fact, but not very clearly. I have mentioned the issue in the comments below.

Another generic comment refers to Tables 1, 2 and where text is centered. This is probably just a formatting oversight, but please make sure the text gets left justified. A related point: make sure column one in table table 2 has separator lines between the items.

More specific comments follow

69: insecurity → security

192: Study characteristics such as socio-economic and demographic → Socio-economic and demographic

277-280: The proportion of acceptable food consumption (35%) in the present study was lower than those reported in Ethiopia (73%) [29] and 86% from a study conducted by the World Food Program and the UN refugee agency UNHCR in Nyarugusa camp in Tanzania, comprising 343 households [30] → The proportion of households with acceptable food consumption reaches 35% in the present study. This is lower that values reported in Ethiopia (73%) [29] and in Nyarugusa refugee camp in Tanzania (86% for a sample of 343 households in a WFP/UNHCR study [30]).

Comment: The current sentence is understandable but nevertheless poorly constructed. For instance “those” refers to (1) [the proportion] reported in Ethiopia and to (2) 86% from a WB study.

286: twice as that → double that

288: could have been due to the fact that the survey was conducted during the short-rainy season (October-December) which is the optimum time for sowing and growing of food products.→ could result from the timing of the survey during the short-rainy season (October-December), the optimum time for sowing and growing of food products.

Even with the suggested change, the sentence remains far too long to be easily understandable. As mentioned at the  beginning of the comments, I also have some doubts about the logic: the low proportion observed is not “because” the survey coincides with the vuli season, but because the vuli season coincides with the lean period. Refer to http://www.fao.org/giews/countrybrief/country.jsp?code=TZA.

329: relatively smaller risk tendency of experiencing severe food insecurity → relatively smaller propensity to severe food insecurity

348: Even though our study was conducted during the short-rainy season

Comment: as mentioned above, this may also have affected your study because it corresponds to the lean period. This is not a problem  per se but it must be kept in mind that the findings correspond to a worst case scenario. Please do mention this in the paragraph that starts on lines 355-356, “The study, however, has limitations.”

396: Why is “United Nations High Commissioner for Refugees” italicized?

p { margin-bottom: 0.25cm; line-height: 120%; background: transparent none repeat scroll 0% 0%; }a:link { color: rgb(0, 0, 128); text-decoration: underline; }a:visited { color: rgb(128, 0, 0); text-decoration: underline; }

Author Response

Please find the responses in the attachment below

Reviewer 3 Report

General comment

The manuscript presents important knowledge on the relationship between social demographics and food access and food security in general. It further proves that larger households will likely suffer from hunger than the smaller household. This is important for any country when implementing economic and transformative policies i.e population control and family/socio-cultural interventions. However, the manuscript still has minor typographical and sentence-making errors, which did not affect the overall scientific output.

Specific/minor comments

Line 17: food insecurity experience scale (FIES)

Change sentence in line 34-35 to “The world population is expected to increase by 2.5 billion between 2007 and 2050, with most of 35 the growth foreseen to occur in urban areas of developing countries

Change sentence in line 60 to “Several studies conducted in developing countries, especially in Africa

Line 176: remove underscore “food_source

Line 214: “Table 3”

Line 230: remove double bracket

Line 232: remove double bracket

Line 235: remove double bracket

Line 238 and 239: remove double bracket

Line 251 to 261: remove double bracket

The statement in line 309 to 310” This is in line with findings from Niger and Nigeria, where the

likelihood of a household being food insecure decreased with an increase in household size [36], does not align with the earlier finding in the study in line 305 to 307 “The fact that in this study, larger households had a significantly higher risk for severe food insecurity than smaller households may be explained by the fact that smaller household sizes are generally better manageable in terms of food demand and supply.

If larger households tend to be more food insure, how does the Niger and Nigerian study be line with the finding of this study where smaller households are more food secure than larger household?

line 305: change the word to "significantly". 

Author Response

(The authors gave the same response as above.)

Round 2

Reviewer 1 Report

While eschewing p-values is a perfectly reasonable choice, this should have been done at the outset prior to statistical analyses and especially not after the original submission reported p-values and claimed evidence for results that were not statistically significant. Changing these decisions after this point runs the risk of readers feeling that the modelling decisions could be driven by the findings.

Even without p-values, I still do not find it reasonable to say things such as (Lines 234–237, but there are several other instances of what I consider to be the same issue in the manuscript): "Polygamous households had a higher chance of severe food insecurity (OR 3.95; 95%CI 1.17; 13.4) [and] also reported [a] higher chance for acceptable food consumption (OR 1.78; 95%CI 0.54; 5.83) compared to those not married." The point estimate for food insecurity has a 95% CI strictly above 1 and so it is entirely reasonable to describe this as "higher" (although this would be a form of significance testing via the CI); however, the point estimate for food consumption extends well below and well above 1, being consistent with half the odds or nearly six times the odds. This cannot be described as "higher" in exactly the same way as the first result and treating these in the same way risks the reader not appreciating the there is evidence for one association in the present study and potential for a (what I would interpret as) practically important association in either direction for the other. If you wanted to say that the point estimate in this case was for 78% higher odds but the data was consistent with between a 46% reduction and a 483% increase in odds, that would be an appropriate interpretation for these results. However, if you do not then do the same for the first result presented above (estimate 295% higher, consistent with 17% to 1240% higher), you will need to include in your statistical methods a comment that for results entirely on one side of the null value, these are interpreted as “higher” or “lower” accordingly, in which case you are back to significance testing via the 95% CI.

The second result above (OR 1.78; 95%CI 0.54; 5.83) simply does not provide any reasonable evidence that the polygamous households differed from those not married in terms of food consumption. It does make the point to me that the study was underpowered as I think that a nearly halving of the odds or more than five-times the odds would both be interesting results, albeit in opposite directions, and so a larger study would be needed to answer this question with greater precision to see which direction this result appears to actually be in. However, this result cannot go far towards answering the question of whether the odds are higher or lower in the non-reference category (if pressed, you would of course guess higher, but you would hardly be surprised if this guess turned out to be incorrect). While the use of point estimates > 1.5 or < 0.67 (I’m assuming this is two-thirds despite the “0.66” on Line 198) might be useful for identifying potentially practically important effects, why base this on the point estimates and not the CI limits if you wish to avoid significance testing? A result with an OR of 1.5 and a 95% CI of 0.75–3.0 is potentially more important (being consistent with an effect of 3.0) than one with an OR of 1.6 and a 95% CI of approximately 1.5–1.7 (where the largest consistent effect is 1.7). Note also that these thresholds will be exceeded more often by chance under the null hypothesis in smaller, underpowered studies for both point estimates and CI limits. Flagging more results as interesting in smaller, underpowered studies is not necessarily a helpful outcome.

As far as I can tell, none of the cited references provided by the authors would support the second result’s interpretation and phrasing from their manuscript. I’m happy to be corrected on this point but at the moment, the approach taken still appears to overstate the findings that I think it would be reasonable to draw from the results.

While you responded that “We previously used the term ‘tendency’ which we derived from the strength of the association, but not from the p-value that can vastly change in different studies, particularly since our study was strongly underpowered. We now present associations based on the point estimate which we specified in the methods section”, the word “tendency” still appears (Lines 303, 317, 320) despite not being defined anywhere in the manuscript (as far as I can tell), as does “significantly” (Line 318, with the same lack of a definition).

My advice would be to focus on where you do have statistically significant evidence (using the level of significance stated in the original manuscript) and interpret the non-statistically significant findings in the Discussion based on their associated 95% CIs in terms of whether this includes values that you would regard as of practical importance. This will ensure that conventionally statistically significant findings are apparent to the reader, as are areas that are or are not worth further investigation. Changing the methods after the fact is not, in my view, appropriate.

I also suggest a careful reading of the manuscript for what could be taken as causal language (e.g. the use of “determine” and it’s inflected forms on Lines 164, 298, 305, and 321).

Author Response

Dear Reviewer 1,

On behalf of my team and I, we would like to thank you for reading our manuscript and for the thoughtful comments and constructive suggestions, which help to improve the quality of this manuscript. 

Please, find attached our responses in  Italics.

Best regard,

Maria Nyangasa
